# Anxiety, Prenatal Attachment, and Depressive Symptoms in Women with Diabetes in Pregnancy

**DOI:** 10.3390/ijerph17020425

**Published:** 2020-01-08

**Authors:** Angela Napoli, Dorian A. Lamis, Isabella Berardelli, Valeria Canzonetta, Salvatore Sarubbi, Elena Rogante, Pietro-Luca Napoli, Gianluca Serafini, Denise Erbuto, Renata Tambelli, Mario Amore, Maurizio Pompili

**Affiliations:** 1Department of Clinical and Molecular Medicine, Sant’Andrea Hospital, Sapienza University of Rome, 00189 Rome, Italy; angela.napoli@uniroma1.it; 2Department of Neurosciences, Mental Health and Sensory Organs, Suicide Prevention Centre, Sant’Andrea Hospital, Sapienza University of Rome, 00189 Rome, Italy; dalamis@gmail.com (D.A.L.); isabella.berardelli@uniroma1.it (I.B.); valeria.canzonetta@gmail.com (V.C.); pietroluca.napoli@gmail.com (P.-L.N.); denise.erbuto@gmail.com (D.E.); 3Department of Psychiatry and Behavioral Sciences, Emory University School of Medicine, Atlanta, GA 30322, USA; 4Department of Psychology, Sapienza University of Rome, 00185 Rome, Italy; salvatore.sarubbi@uniroma1.it (S.S.); elena.rogante@gmail.com (E.R.); 5Department of Neuroscience, Rehabilitation, Ophthalmology, Genetics, Maternal and Child Health (DINOGMI), Section of Psychiatry, University of Genoa, IRCCS Ospedale Policlinico San Martino, 16132 Genoa, Italy; gianluca.serafini@unige.it (G.S.); mario.amore@unige.it (M.A.); 6Department of Dynamic and Clinical Psychology, Sapienza University of Rome, 00185 Rome, Italy; renata.tambelli@uniroma1.it

**Keywords:** diabetes in pregnancy, prenatal attachment, depressive symptoms, anxiety

## Abstract

The purpose of this study was to evaluate the relationship between anxiety, prenatal attachment, and depressive symptoms among women with diabetes in pregnancy. Participants were 131 consecutive pregnant women between the ages of 20 and 45 with a diagnosis of gestational or pregestational type 1 or type 2 diabetes. Data on previous psychiatric symptoms were obtained from the Anamnestic and Social Questionnaire and the Mini-International Neuropsychiatric Interview (MINI). Information on prenatal attachment was collected using The Prenatal Attachment Inventory (PAI), and The Edinburgh Postnatal Depression Scale (EPDS) assessed depressive symptoms in the third trimester of pregnancy (at a mean of 25 weeks). Results demonstrated that in women affected by diabetes in pregnancy, two facets of prenatal attachment (anticipation, interaction) were negatively correlated with depressive symptoms, and a history of anxiety, assessed with the MINI, moderated the relation between the prenatal attachment interaction factor and depressive symptoms during pregnancy.

## 1. Introduction

Pregnancy is one of life’s major events, which leads to social, psychological, and hormonal changes and may contribute to the development of mental disorders, including anxiety and depressive disorders. Recent studies have found an association between diabetes and depression among non-pregnant patients, highlighting how diabetes may increase the risk for depression [1] as well as how depression may be a risk factor for type 2 diabetes [2]. In addition, some researchers have suggested that both conditions may share common biological mechanisms [3]. Although it has been shown that gestational diabetes mellitus (GDM)—a form of diabetes occurring only in pregnancy [4]—is an established risk factor for diabetes [5], evidence for the relation between diabetes in pregnancy and depression during pregnancy or postpartum is limited [6]. A recent review demonstrated that between 10% and 20% of women experience depressive symptoms during pregnancy [7] and among women with GDM, the prevalence of depression during or after pregnancy ranged widely, from 4.1% to 80% [8]. Previous studies demonstrated that a history of anxiety disorders prior to pregnancy is related to antenatal depressive symptoms [9,10]. Susceptibility to depressive symptoms during pregnancy is possibly linked to maternal cortisol levels [11] and pregnant women with diabetes appear to be at higher risk of developing depressive symptoms. Depressive symptoms can develop directly from hyperglycemia (related to increased oxidative stress, inflammation, or leptin resistance induced by hyperglycemia), and indirectly from psychological stress related to the diagnosis of diabetes [12,13].

Over the last few years, in the wake of Bowlby’s empirical study of human attachment, various theories have been developed related to the experience of pregnancy [14]. Deutch, Bibring and Benedeck described prenatal attachment as a process in which a pregnant woman’s psychic energy is emotionally invested into the fetus [15,16,17,18]. This early relationship has received growing scientific interest [19] that has led to the description of a formal theory of prenatal attachment. Muller [20] redefined prenatal attachment as “the unique relationship that develops between a woman and her fetus” (p. 11). These feelings are not dependent on the feelings the woman has about herself as a pregnant person or her perception of herself as a mother [21,22]. Studies have demonstrated individual differences varying from being highly attached early in the pregnancy, to low, or no, attachment during the pregnancy [23]. Recent research has demonstrated that cognitive, emotional, and situational factors are associated with the level of prenatal attachment [24,25]. Among these factors, remarkable importance has been attributed to social support [26], twin pregnancies [27], loss or stillbirth in a previous pregnancy [28], maternal age [29], maternal personality [30], physical symptoms, body image [31,32], as well as depression and anxiety [33,34]. However, the relation between anxiety [35], depression [36,37], and prenatal attachment is not well clarified.

On the basis of existing literature [38] and consistent with theory [39], we hypothesized that in women diagnosed with pregestational diabetes and GDM, (1) the three facets of prenatal attachment (anticipation, differentiation, interaction) would be negatively associated with depression and (2) a history of anxiety would moderate the relation between the significant prenatal attachment facets and depression during pregnancy.

## 2. Materials and Methods

### 2.1. Participants

Participants were recruited during their first visit to the Endocrinology outpatients at the Diabetes and Pregnancy Clinic at Sant’Andrea Hospital. The patients during their pregnancy sought consultation from the outpatient clinic of Diabetes and Pregnancy, a branch of the internal medicine unit, as they presented impaired blood glucose curves or because they already had a diagnosis of type 1 or 2 diabetes. Inclusion criteria were any women with a diagnosis of gestational diabetes or with pregestational diabetes, in pregnancy. Exclusion criteria involved major chronic diseases such as cardiovascular diseases, severe autoimmune diseases (rheumatic arthritis, sclerodermia, severe psoriasis, etc.), neurological diseases (multiple sclerosis, infectious disease (HIV, viral hepatitis, etc.), psychosis, and congenital malformations. The time period for data collection was twelve months. Participants were 131 consecutive pregnant women between the ages of 20 and 45 (mean = 34.29, SD = 5.32) with a diagnosis of gestational or pregestational type 1 or type 2 diabetes. The majority had gestational diabetes (*n* = 109, 83.2%), whereas the remaining women were diagnosed with type 1 or 2 diabetes prior to pregnancy (*n* = 21, 16.2%). Of the participants, 85 (64.4%) were married and 45 (34.6%) were not. Moreover, 19 (14.5%) women reported having a history of depression, whereas 46 (35.1%) reported a history of anxiety. The medical staff informed patients about the existence of this study and all patients signed informed consent. All patients participated voluntarily and the study was approved by the institutional review board. All demographic and clinical variables are shown in Table 1.

### 2.2. Measures

The Mini-International Neuropsychiatric Interview (MINI) is a clinically administered tool used in our environment to assess both psychiatric and non-psychiatric patients and rule out the presence of any serious psychiatric disorder. Consolidated experience in the assessment of non-psychiatric patients indicated that anxiety is the most common non-invalidating psychiatric condition traceable in such populations. Appropriately trained clinicians using the MINI assessed the diagnosis of generalized anxiety. The MINI is a short-structured interview with high validity and reliability [40]. Although the MINI is not a substitute for a psychiatric clinical interview, validation studies have confirmed the validity of this instrument as a reliable tool in psychiatry [41,42]. Other more specific constructs for assessing patients enrolled in this study were the Anamnestic and Social Questionnaire, the Prenatal Attachment Inventory (PAI) and the Edinburgh Postnatal Depression Scale (EPDS). Assessment of pregnant women was performed by trained psychologists and supervised by two fully qualified psychiatrists. All of them were administered during the 25th week of pregnancy.

The Anamnestic and Social Questionnaire is a questionnaire developed for this study in our Department of Neurosciences, Mental Health and Sensory Organs at Sant’Andrea Hospital, Sapienza University of Rome, Italy. The Anamnestic and Social Questionnaire is an instrument administered by the principal investigator and consists of 35 questions, most of which are multiple-choice and some are open-ended questions. It was developed to obtain an available and manageable instrument that could simultaneously investigate many medical and social aspects related to pregnancy. The first part of this anamnestic instrument provided basic information about the pregnant woman including age, marital status, level of education, and occupation and information about the pregnancy and the family. The second part included questions concerning the diagnosis of diabetes, the third part was dedicated to the pregnancy, and the fourth section included questions on mental health conditions of the pregnant women both in the present and in the past and the medical condition during the pregnancy. In particular, four questions investigated the presence of anxiety and depression disorders before pregnancy; possible therapeutic (pharmacological and/or psychological) therapies and familiarity were investigated.

The Prenatal Attachment Inventory (PAI) [43,44] is a self-report questionnaire designed to measure prenatal attachment in terms of the unique affectionate relationship that develops between a mother and her fetus. As recommended by a factor analysis conducted by Pallant et al. [45], three six-item PAI subscales (Anticipation, Interaction, Differentiation) were used in the current analyses. Sample items included statements such as “I imagine calling the baby by name” (anticipation), “I think that my baby already has a personality” (differentiation), and “I stroke the baby through my tummy” (interaction). The response options range from 1 = almost never to 4 = almost always, and scores on each subscale range from 6 to 24, with higher scores indicating higher levels of prenatal attachment. The PAI has been shown to have strong psychometric properties [46] and has been successfully used in Italian women [43,44,45,46,47]. In the current sample, the internal consistency reliability estimates were 0.72, 0.67, and 0.72 for the anticipation, differentiation, and interaction subscales, respectively.

The Edinburgh Postnatal Depression Scale (EPDS) [48] is a self-administered 10-item questionnaire designed to assess postnatal depressive symptoms in women. It is one of the most widely used questionnaires in various countries for the screening of postnatal depressive symptomatology [49]. According to the results obtained by Murray and Cox [50], who highlighted that EPDS was effective in identifying women with major depression (level II evidence) during pregnancy, we used the EPDS in this study for the assessment of depression during pregnancy. The questionnaire items investigate the presence and intensity of depressive symptoms during the previous seven days, specifically anhedonia, self-blame, anxiety, fear or panic, inability to cope, difficulty in sleeping, sadness, tearfulness, and thoughts of self-harm. Sample items include “I have been so unhappy that I have been crying” and “I have been so unhappy that I have had difficulty sleeping.” The items are scored from 0 to 3, increasing according to the severity of the symptoms, and a summed score ranging from 0 to 30. The EPDS has demonstrated strong validity and reliability [51,52], and the Italian version of the EPDS has been used in previous studies of women post-delivery [53,54]. According to previous studies, we considered a cut-off score of 13 for the diagnosis of depression in pregnancy [55]. In the current sample, the reliability estimate was 0.78.

### 2.3. Statistical Analyses

In order to determine the sample size for a moderation analysis, a power analysis was conducted using G*Power [56] (Institute for Experimental Psychology in Dusseldorf, Dusseldorf, Germany). The analysis was based on a hierarchical linear regression, with a medium effect size (f2) of 0.10, an alpha of 0.05, a standard power level of 0.80, a total of four tested predictors, and eight total predictors. The results of the power analysis showed that a minimum of 125 participants would be needed to achieve an appropriate power level for this study. Accordingly, our sample size of 131 was deemed appropriate to detect main and interaction effects. In order to allow our four predictors to compete with each other, above and beyond the effects of the covariates, we conducted a series of hierarchical regressions. The covariates were entered in the first step, history of anxiety was entered in the second step, and the third step consisted of the three PAI subscales. In the fourth step, we tested the interaction effect of a history of anxiety with the interaction subscale of the PAI on depressive symptoms. As recommended for testing moderation effects, all predictor and interaction terms were centered prior to model estimation to reduce multicollinearity and improve interpretation of regression coefficients [57,58].

## 3. Results

Sociodemographic and clinical variables are shown in Table 1.

All partial correlations among the primary study variables are shown in Table 2. We did not find any significant association between psychological, physical, and socioeconomic maternal variables and depression during pregnancy.

Results demonstrated that a history of anxiety was associated with depressive symptoms during pregnancy (r = 0.20, *p* = 0.044) after controlling for diabetes type, age, marital status, and past history of depression. In line with our hypothesis, the anticipation (r = −0.24, *p* = 0.02) and interaction (r = −0.34, *p* = 0.001) subscales of the PAI were related to depressive symptoms; however, the differentiation facet was not correlated with depressive symptoms (r = −0.16, *p* = 0.13). Total correlation showed that age was related with anticipation (r = −0.22, *p* = 0.01) and interaction (r = −0.11, *p* = 0.03) subscales of the PAI. Moreover, not being married was associated with a history of anxiety (r = 0.22, *p* = 0.01). To further investigate these associations, a hierarchical regression analysis was conducted.

In Model 1, we first entered patient type (gestational diabetes onset in pregnancy vs. type 1 or 2 onsets prior to pregnancy), age, marital status, and history of depression as covariates, which accounted for 15.5% of the total variance, ΔF(4, 126) = 5.77, *p* < 0.001. In Model 2, we entered the history of anxiety in the second step, which predicted depressive symptoms during pregnancy above and beyond the covariates (B = 1.63, *p* = 0.043) and accounted for an additional 2.7% of the total variance ΔF(1, 125) = 4.19, *p* = 0.043. In Model 3, we entered anticipation, differentiation, and interaction in the third step as potential predictors of depressive symptoms during pregnancy. Results indicated that interaction with the fetus was a significant predictor (B = −0.41, *p* = 0.018) over and above the covariates and a history of anxiety, accounting for an additional 6.9% of the total variance (25.1%), ΔF(3, 122) = 3.71, *p* = 0.013; however, anticipation (B = −0.14, *p* = 0.32) and differentiation (B = 0.14, *p* = 0.35) were not associated with depressive symptoms. Given that anticipation and differentiation were not found to significantly predict depressive symptoms, these variables were omitted from subsequent analyses.

We entered the interaction term (i.e., anxiety X PAI interaction) into the regression analysis as a fourth step, preceded by the covariates (in the first step), history of anxiety (in the second step), and the three main effects of the PAI subscales (in the third step) for testing if PAI Interaction in women with history of anxiety predicts depressive symptoms. As can be seen in Table 3, our results revealed that the interaction of history of anxiety and interaction with the fetus (B = 0.65, SE = 0.27, *p* = 0.016; Model 4) was a significant predictor, which accounted for an additional 3.5% of the variance in depressive symptoms, with 28.6% total variance explained, ΔF(1, 121) = 5.99, *p* = 0.016.

To facilitate interpretation, the regression lines for interaction with the fetus (Figure 1) with depressive symptoms were plotted for the two anxiety groups (no history vs. history) as recommended by Aguinis et al. [59]. As can be seen in Figure 1, the slope was significantly steeper for the no anxiety group (r = −0.28, *p* = 0.02) compared to the history of the anxiety group (r = 0.08, *p* = 0.64). Instead, no relation was found with PAI Interaction and depressive symptoms in the groups with and without a history of depression.

## 4. Discussion

In the current study, our results indicated that two facets of prenatal attachment (anticipation, interaction) were negatively correlated with depressive symptoms in women with gestational diabetes, and a history of anxiety moderated the relation between the prenatal attachment interaction factor and depressive symptoms. Our results confirmed previous studies that demonstrated the relation between high anxiety levels and depression during pregnancy [60]. Many studies have investigated the main risk factors for antenatal anxiety and have highlighted a complex multi-factorial etiology [61,62]. It has been demonstrated that women who experienced antenatal anxiety were about three times more likely to suffer from depression during pregnancy [63]. Several psychological and social risk factors for antenatal depression have been well described, including early age, low income, lower educational attainment, a history of depression, a history of miscarriage and pregnancy termination, a history of childhood sexual abuse, and low social support [64,65,66,67]. Research suggests that depression during pregnancy is associated with an increased risk for emotional [68], behavioral [69], and cognitive problems of offspring [70], possibly due to elevations in maternal cortisol levels [71]. Although women with diabetes in pregnancy have a greater risk of developing depressive symptoms, studies of depression in women with GDM are scarce in the literature, and most of them have evaluated women diagnosed with pre-pregnancy diabetes or postpartum depression [72].

Consistent with these findings, our study demonstrated that a history of anxiety was associated with prenatal depressive symptoms in women affected by diabetes in pregnancy. Assuming that high levels of pregnancy-related stress and/or other stressors can activate a woman’s coping strategy [73], we hypothesized that pregnant woman with a past diagnosis of anxiety disorders utilized less adaptive strategies for coping with pregnancy-related stress and diabetes-related stress, and this is associated with higher depressive symptoms during pregnancy. The increased glucose levels in women affected by diabetes in pregnancy can cause direct and indirect maternal distress, and this factor becomes more complicated if the woman is exposed to environmental stressors [74]. McNamara et al., in a recent review on the relation between maternal wellbeing and maternal fetal attachment, observed that depression in the antenatal periods was associated with lower prenatal attachment. Moreover, the closer the prenatal attachment of a mother to her unborn child, the fewer symptoms of depression she reports during the last term of pregnancy and postpartum, demonstrating that maternal mood may negatively impact a mother’s ability to bond with her baby during pregnancy [75]. Our result highlights the importance of a previous history of anxiety disorder in moderating the relation between prenatal attachment and depressive symptoms in GDM patients. The relationship between prenatal attachment and psycho-affective factors in pregnant women has been established by Ossa et al. [76]. Specifically, the researchers observed the association of poorer prenatal attachment and unwanted pregnancy, higher levels of perceived stress, prenatal depression, and low family support [37,38,39,40,41,42,43,44,45,46,47,48,49,50,51,52,53,54,55,56,57,58,59,60,61,62,63,64,65,66,67,68,69,70,71,72,73,74,75,76,77]. According to previous studies reporting that the most important determinant of prenatal attachment is the mother’s mental health, [36,37,38,39,40,41,42,43,44,45,46,47,48,49,50,51,52,53,54,55,56,57,58,59,60,61,62,63,64,65,66,67,68,69,70,71,72,73,74,75,76,77,78,79], our results stressed that prenatal attachment was reduced when depressive symptoms were high; however, no other trials considered a previous anxiety disorder as a moderator of this relationship. Thus, the results from our study expand prior findings by suggesting that pregnant women who have experienced anxiety in the past and who have a decreased amount of interaction with their fetus may be at an elevated risk of reporting depressive symptomatology, and, those women who exhibited depressive symptoms had alterations in PAI subscales. Specifically, the PAI “interaction” subscale was associated with prenatal depressive symptoms strongly, as compared to other subclasses, stressing that depressive symptoms are associated with a mother’s lesser feelings for the fetus and with a difficulty of sharing her experience with others. These results also confirmed that the three PAI-R subscales should be kept separate, and not summed as a total score, as they assessed different aspects of the prenatal attachment construct [80].

One of the limitations of this study is that we collected data mainly in the third trimester (at a mean of 25 weeks) and not in other gestational trimesters or before delivery because the gestational diabetes is generally screened and diagnosed after the 24th gestational week. Second, it was not possible to make comparisons with pregnant women without diabetes since all of the women included in the study had GDM. Third, we did not assess the temporal relationship between the onset of depressive symptoms in pregnancy and prenatal attachment; therefore, we cannot determine whether the alterations in prenatal attachment were due to depressive symptoms or are a separate construct. Future research should assess anxiety symptoms with a valid and reliable instrument, such as the Perinatal Anxiety Screening Scale. Fourth, diagnostic interviews conducted by clinicians were not used, and thus, depressive symptoms were measured using the EPDS. Although the EPDS is a valid and reliable instrument, it assesses depressive symptoms, not clinical diagnoses of depression. Thus, clarification is needed regarding the role of major depressive disorder versus depressive symptoms in the relations examined in the current study.

Another issue that merits discussion is the social desirability bias with the use of self-report instruments and the cross-sectional nature of the study that does not follow individuals over time to determine cause and effect.

## 5. Conclusions

Based on the results of this study, assessing a previous anxiety disorder with a short-structured interview as the MINI and assessing attachment during pregnancy using the PAI should aid in identifying women with diabetes during pregnancy who suffer from anxiety and report lower levels of attachment. This would allow clinicians to identify women at risk of developing depression and allow us to follow women at risk of developing depression during their pregnancy. Furthermore, the importance of assessing factors related to perinatal depression is supported by recent studies, which demonstrated an association between prenatal maternal depressive symptoms and brain development of offspring. Several studies highlighted the relationship between exposure to maternal prenatal stress, and physical, psychological, and psychiatric outcomes in later life, including emotional, behavioral, and cognitive psychopathology, stress physiology, brain plasticity, immune function, and chronic metabolic diseases. Exposure to elevated levels of maternal glucocorticoids can create persistent changes in fetal biological systems, increasing the risk for developmental disorders later in life [80].

In conclusion, inquiring about the prenatal attachment, which will ultimately develop into the mother–child bond after delivery, may be of clinical importance. Specific interventions during pregnancy such as psychotherapy should not only address the depressive symptoms but could also promote the well-being of mother and child by working on the promotion of prenatal attachment.

## Figures and Tables

**Figure 1 ijerph-17-00425-f001:**
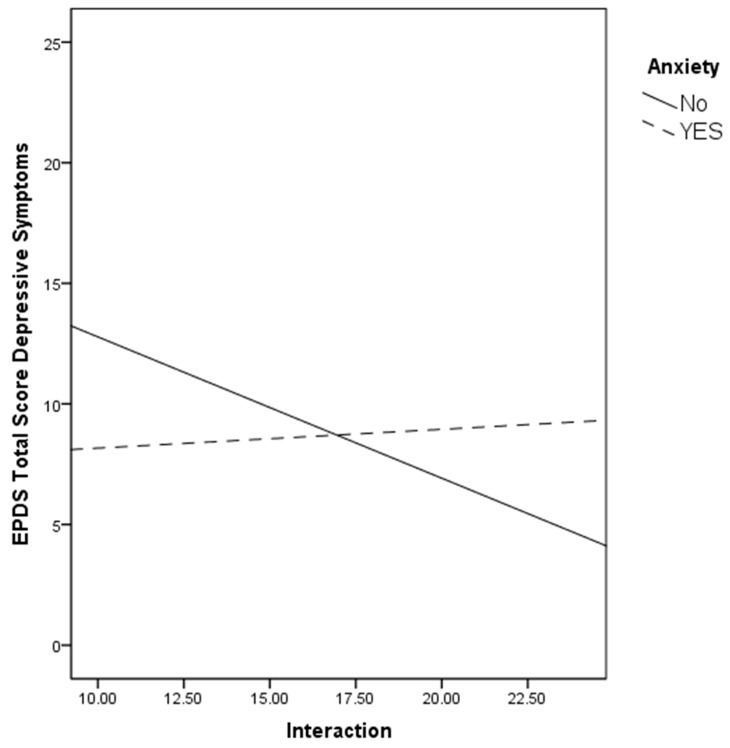
The relation between the PAI subscale and depressive symptoms by women’s history of anxiety.

**Table 1 ijerph-17-00425-t001:** Sociodemographic and clinical variables.

Variables		*N*	%
Patient		131	100
Age M ± SD	Range 20–45	M = 34.29 ± 5.32	
Week of pregnancy M ± SD	Range 6–38	M = 24.59 ± 8.73	
Marital status	Married	85	64.4
	Not Married	45	34.6
Diagnosis	Gestational Diabetes	109	83.2
	Pregestational type 1 or type 2	21	16.2
Psychological disorder history	History of anxiety	46	35.1
	History of depression	19	14.5
	History of both anxiety and depression	14	10.7

**Table 2 ijerph-17-00425-t002:** Correlation matrix, means, and standard deviations (SD) among Anxiety history, Prenatal Attachment subscales and Depressive symptoms, Age, and Marital status.

Variables	1	2	3	4	5	6	7
1. Anxiety history	--						
2. Anticipation	0.11	--					
3. Differentiation	0.02	0.62 **	--				
4. Interaction	0.01	0.60 **	0.66 **	--			
5. Depressive symptoms	0.20 *	−0.24 *	−0.16	−0.34 **	--		
6. Age	0.05	−0.22 *	−0.11	−0.19 *	0.01	--	
7. Marital status	0.22 *	0.04	0.01	0.10	0.17	0.05	--
Mean	0.35	17.87	16.89	19.53	7.84	34.29	1.34
SD	0.48	3.61	3.57	3.08	4.75	5.32	0.46

Note. *N* = 131. Tabled values are partial correlations controlling for patient type, age, marital status, and history of depression. * *p* < 0.05; ** *p* < 0.01.

**Table 3 ijerph-17-00425-t003:** Summary of the hierarchical regression analysis for Patient type, Age, Marital status, Depression history, Anxiety history and Prenatal Attachment subscales predicting women’s depressive symptoms.

	Model 1			Model 2			Model 3			Model 4		
Variables	B	SE	β	B	SE	β	B	SE	Β	B	SE	β
Patient type	−1.66	0.96	−0.14	−1.55	0.95	−0.13	−1.80	0.95	−0.15	−1.78	0.93	−0.12
Age	−0.02	0.07	−0.02	−0.02	0.07	−0.02	−0.08	0.07	−0.09	−0.08	0.67	−0.09
Marital status	0.97	0.77	0.10	0.68	0.78	0.07	0.92	0.76	0.10	1.07	0.75	0.12
Depression history	4.36	1.04	0.35 **	3.68	1.08	0.30 **	2.96	1.06	0.24 **	2.87	1.04	0.23 *
Anxiety history				1.63	0.80	0.18 *	1.72	0.78	0.19 *	1.77	0.76	0.19 *
Anticipation							−0.14	0.14	−0.11	−0.11	0.14	−0.08
Differentiation							0.14	0.15	0.10	0.13	0.15	0.10
Interaction							−0.41	0.17	−0.26 *	−0.61	0.19	−0.40 **
Anxiety X Interaction										0.65	0.27	0.22 *
R^2^	0.16			0.18			0.25			0.29		
F for change in R^2^	5.77 **			4.19 *			3.71 *			5.99 *		

Note. *N* = 131. * *p* < 0.05; ** *p* < 0.01.

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
