# Peer review of "Anxiety, Prenatal Attachment, and Depressive Symptoms in Women with Diabetes in Pregnancy"

_ijerph, 2020, doi:10.3390/ijerph17020425_

Round 1

Reviewer 1 Report

I found the study to be interesting and helpful regarding the associations of anxiety, depression and prenatal attachment for pregnant women with diabetes.

I would liked to have heard more in the conclusion regarding identifying pregnant women with diabetes and their at-risk status for depression, and follow-up (the authors last statement in the conclusion is very interesting and I would like more information in the manuscript regarding prenatal maternal depressive symptoms and brain development in offspring.)

The authors should proofread the article as there were some minor grammatical errors and/or phraseology.  E.g., in the conclusion section, first paragraph:

  Based on the results of this study, assessing attachment during pregnancy using the PAI, identifies those women with diabetes during pregnancy who suffered from anxiety, and presented lower levels of attachment.  (Take out: provides to identifying).

Overall, an excellent study.

Reviewer 2 Report

This is an interesting article examining the relationship between prenatal attachment and depressive symptoms in pregnant diabetic women. So, there are some problems with the description method that need to be improved before publication. In addition, I hope the authors provide more detailed data so that readers can understand the conclusions clearly.

Abstract

The conclusion “Based on the results of this study…”does not match the purpose of this study, and please make any necessary corrections. Please note that this data is a cross-sectional analysis and whether PAI can predict future development of depression has not been determined.

Introduction

Final paragraph; Multiple aims are written and readers may be confused. Please present the main one aim that is consistent with the aim of the abstract. Then, in the next section “2. Hypothesis”, please describe the three hypotheses you want to prove to reach that goal of this study.

Material and Methods

1) Please give each paragraph a proper heading. (3.1 Participants; 3.2. assessment of …).

2) Participants; It is easy to understand the demographic factors of the participants when they are shown in one table. How many women had both a history of anxiety and depression?

3) Subsection “Statistics” should be placed in the last of “Material and Methods” section. Explanation on Figure 1 (line181-183) should be placed in the “Results” section.

Results

1) Tables and a figure should be presented in a way that is self-explanatory. Therefore, please give a more detailed title to each one.

2) The reader may be interested in the direct relationship between various background factors (age, marital status, level of education, patient type, etc) and PAI or EPDS. So, if possible, please add these correlations to Table 1.

3) For most readers, it is difficult to understand the story of the fourth step of this regression analysis. Please explain more clearly about how to calculate “Anxiety X Interaction” for each woman, and why this result led to “Figure 1”

4) In relation to Figure 1, please present correlations between EPDS and Interaction in women with or without a history of anxiety separately. It is interesting if the same can be said in the history of depression.

Discussion

1) Line 10-23 of the first paragraph and the second paragraph was the discussion on the relationships between a history of anxiety and prenatal depressive symptoms and thus please put them into one paragraph.

2) Please add one paragraph that discusses the relation between prenatal attachment and prenatal depressive symptoms. Please discuss why PAI subscales ”interaction” was associated with prenatal depressive symptoms so strongly, as compared to other subclasses.

Conclusions

It seems that the conclusions based on the results of this study are not accurately described.
